# Progress toward Health System Readiness for Genome-Based Testing in Canada

Don Husereau [1,*] , Eva Villalba [2] , Vivek Muthu [3] , Michael Mengel [4] , Craig Ivany [5] , Lotte Steuten [6,7] , Daryl S. Spinner [8] , Brandon Sheffield [9] , Stephen Yip [10] , Philip Jacobs [11] , Terrence Sullivan [12,13] and Larry Arshoff [14]

1   School of Epidemiology and Public Health, University of Ottawa, Ottawa, ON K1G 5Z3, Canada
2   Coalition Priorité Cancer au Québec, Saint-Lambert, QC J4P 2J7, Canada; evav@coalitioncancer.com
3   Marivek Healthcare Consulting, Epsom KT18 7PF, UK
4   Department of Laboratory Medicine & Pathology, University of Alberta, Edmonton, AB T6G 2S2, Canada
5   Provincial Health Services Authority, Vancouver, British Columbia, Vancouver, BC V5Z 1G1, Canada
6   Office of Health Economics, London SE1 2HB, UK
7   City University of London, London EC1V 0HB, UK
8   Menarini Silicon Biosystems Inc., Huntingdon Valley, PA 19006, USA
9   William Osler Health System, Brampton, ON L6R 3J7, Canada
10  Department of Pathology & Laboratory Medicine, Faculty of Medicine, University of British Columbia, Vancouver, BC V6T 1Z7, Canada
11  Faculty of Medicine and Dentistry, University of Alberta, Edmonton, AB T6G 2R3, Canada; philjacobs@shaw.ca
12  Institute of Health Policy, Management and Evaluation, University of Toronto, Toronto, ON M5T 3M6, Canada
13  Gerald Bronfman Department of Oncology, McGill University, Montreal, QC H4A 3T2, Canada
14  Diagnosis, Solutions & Results Inc., Thornhill, ON L4J 7N5, Canada
*   Correspondence: dhuserea@uottawa.ca; Tel.: +1-613-299-4379

**Abstract:** (1) Background: Genomic medicine harbors the real potential to improve the health and healthcare journey of patients, care provider experiences, and improve the health system efficiency—even reducing healthcare costs. There is expected to be an exponential growth in medically necessary new genome-based tests and test approaches in the coming years. Testing can also create scientific research and commercial opportunities beyond healthcare decision making. The purpose of this research is to generate a better understanding of Canada's state of readiness for genomic medicine, and to provide some insights for other healthcare systems. (2) Methods: A mixed-methods approach of a review of the literature and key informant interviews with a purposive sample of experts was used. The health system readiness was assessed using a previously published set of conditions. (3) Results: Canada has created some of the established conditions, but further action needs to be taken to improve the state of readiness for genome-based medicine. The important gaps to be filled are the need for linked information systems and data integration; evaluative processes that are timely and transparent; navigational tools for care providers; dedicated funding to facilitate rapid onboarding and support test development and proficiency testing; and broader engagement with innovation stakeholders beyond care providers and patients. These findings highlight the role of the organizational context, social influence, and other factors that are known to affect the diffusion of innovation within health systems.

**Keywords:** diagnostic molecular pathology; genetic testing; diagnostic services; technology assessment; biomedical; genetic services; financial support; clinical governance; health technology; healthcare innovation



## 1. Introduction

The rapidly developing field of genomic medicine and genome-based testing has led to an increasing number of administrative decisions regarding the implementation of new

testing. These include decisions about different technical platforms (e.g., single-gene, multi-gene, whole-exome, and whole-genome sequencing and expression analysis); modalities (tissue, saliva, blood, cerebrospinal fluid, or urine-based sampling); location (laboratory-based or delivered at point-of-care); provenance (commercially available in vitro diagnostic tests and services versus in-house/laboratory-developed tests); and the timing and sequencing of tests. All of these factors affect how patients and health systems may benefit from genomic medicine ('clinical utility'), including cost impacts and patient outcomes, and broader health system goals such as caregiver and patient experiences [1]. They also influence how care is delivered.

A significant challenge with integrating genome-based testing into the healthcare environment is the need for revisiting the role of traditional laboratory services and care pathways [2,3]. Testing is a complex intervention [4], and the readiness to implement new testing requires adequate infrastructure, as well as operational and broader healthcare conditions [5] in order to be fully prepared to deliver tests effectively, efficiently, and without jeopardizing timeliness in patient care. Attention to these conditions can help policymakers achieve healthcare goals, including timely and equitable access to testing; providing efficient high-value care; avoiding unnecessary duplication of services, care interruptions, and wait times; appropriate management of personal information; and improving the patient and caregiver journey (Table 1).

**Table 1.** Necessary conditions for healthcare system preparedness for genome-based testing from reference [5].

| Health System Challenge | Condition(s) Required | Good Practice Description |
|---|---|---|
| Care interruptions, wait times, and unsustainable care | (1) Resource planning | Frequent and systematic resource planning |
| | (2) Finance model | Nimble, value-based funding formula (i.e., payment that accounts for development and human resource costs and benefits) |
| Inequitable care delivery | (3) Creating communities of practice and healthcare system networks | Engagement with all stakeholders, including administrators, information and communications technology professionals, implementation and genome scientists, and public and private sector innovators |
| Uncoordinated or duplicative care | (4) Informatics | Integrated laboratory information systems and electronic health records |
| | (5) Tools for awareness and care navigation | Available, up-to-date information and navigation support |
| | (6) Tools for education and training | Educational standards that address continuing professional development, knowledge updates, and transfer and quality improvement |
| Technology creep or failure to keep up with pace of innovation | (7) Single entry/exit point for innovation proposals | Application process open to all stakeholders with explicit timelines |
| | (8) Integration of innovation and healthcare delivery | Integration of future testing; private/public sector partnerships |
| Inequitable or inefficient care | (9) Evaluative function | Consistent and adherent to key HTA principles such as timeliness and transparency |
| | (10) Service model | System-wide care coordination |
| Legal liability, low care quality | (11) Regulation | System-wide analytic standards and regulation that addresses human resource qualifications and training, documentation of records, quality control processes, and proficiency testing |
| | (12) Data privacy and security | System-wide privacy standards |

In addition to achieving these healthcare goals, genomic medicine also creates opportunities for scientific discovery and economic growth [6]. Already, some countries have created substantial investment in overhauling their genomic medicine services. The NHS England Genomic Medicine Service, for example, was built on a commitment to provide consistent and equitable care, common national standards, a single national test directory, and to give patients opportunities to participate in research, while building a genomic knowledge base to inform innovation [7].

Unlike the UK's centralized healthcare system, Canada's provinces and territories are individually responsible for genome-based testing. Within each province, the capacity for advanced diagnostic testing is typically concentrated in larger specialized tertiary care centres that provide essential healthcare services but are owned and operated independently of the government. As such, there is no 'Canadian' genetic testing service. Each Canadian healthcare jurisdiction must individually prepare itself for a future of genomic medicine.

Their ability to do so will, in turn, vary by the individual healthcare system organization and structure, resource availability, geography, and other factors. While each region is making some progress toward the goal of readiness, little is known about the overall progress in Canada, including a comprehensive understanding of the barriers of Canada-wide readiness and the potential solutions to remedy these barriers. The purpose of this research is to generate a better understanding of Canada's progress, and to provide some insights into the possible barriers and facilitators of readiness for other healthcare systems.

## 2. Materials and Methods

Five Canadian regions (Ontario, Quebec, British Columbia, Alberta, and Nova Scotia), representing almost 90% of Canada's population, were chosen for this report. The current state of each healthcare system was informed using a mixed-methods approach. A narrative review of the literature was conducted based on a purposive sample of the commercially published and grey literature including health ministry and healthcare system websites. In parallel, information was sought using a conventional content approach and based on semi-structured interviews ($n = 39$; 30–60 min) with key informants. The interviews were performed from an approach of qualitative description [8], that is, a naturalistic inquiry where the interviews and resulting data present 'a rich, straight description of an experience or event' [9]. All interviews were conducted by D.H. with a purposive sample of experts including regional representatives from the commercial life science sector (6 pharmaceutical ($n = 20$) and 2 diagnostic ($n = 6$) companies); academia/research ($n = 3$); healthcare providers ($n = 3$); healthcare leadership/administrators ($n = 6$); and patients ($n = 1$). Informants were chosen based on differing expertise and geographic location with some ($n = 4$) having previously worked with the author.

Interviews were conducted via a recorded video conference call with an audio recording feature and transcription capability. An interview guide (see Appendix A) was developed and pilot tested, with all but 2 participants (1 healthcare provider and 1 researcher) who were approached agreeing to be interviewed. Participants were provided summary notes from transcripts for verification (member checking). Data, including all audio recordings, transcripts, and digital field notes, were stored on a password-protected drive. Automated transcriptions were checked against the audio recording and subsequently corrected. Interviewees were asked to provide their perspectives from an organizational perspective without risk of personal injury; therefore, no ethics approval was obtained.

Information obtained from the literature search and interviews was then compiled and compared against 11 of 12 conditions for health system readiness previously published [5]. One condition (data privacy) was not applied as it is addressed by overarching federal legislation and not believed to differ across regions. The state of each condition for each jurisdiction was then judged to either 'needs improvement' (i.e., non-existent or nascent); 'partially established' (i.e., some components of the condition established); or 'established' (most to all components of the condition established).

## 3. Results

*3.1. Canada*

Overall, Canada appears to be making progress, but is only partially ready for a future of genomic medicine. The important gaps to be filled are the need for linked information systems and data integration (informatics); evaluative processes that adhere to health technology assessment (HTA) principles of timeliness transparency and engagement; navigational tools for care providers; dedicated funding to facilitate rapid onboarding or a funding formula (i.e., value-based payment model for the reimbursement of test services) that supports test development and proficiency testing; and broader engagement with a broader set of innovation stakeholders (e.g., patients, administrators, information and communication technology professionals, implementation and genome scientists, public and private sector innovators, and others).

The Canadian provinces that are in a better state of readiness for genomic medicine are Alberta and Quebec. This is largely due in part to the earlier establishment of single laboratory service organizations and programs that provide the necessary infrastructure for coordination and planning as well as necessary operational conditions (Table 2).

**Table 2.** State of readiness for genome-based testing in Canada. Strengths and weaknesses of individual provinces.

| Province | Population (M)/Area (1000 km$^2$) | Testing Centres * | Strengths | Weaknesses | Rank ** |
|---|---|---|---|---|---|
| Alberta | 4.3/661.8 | 5 | • Single service organization (Alberta Precision Laboratories) that provides oversight and resource planning. <br>• Integration of laboratory information across province is established. <br>• Integration and exchange with innovators through dedicated translational research programs and mainstream use of investigational testing. | • The test review process, timelines, and criteria used are not publicly available. <br>• There are still opportunities to improve test navigation and educational standards for patients and providers. | 1 |
| Quebec | 8.5/1542.0 | 7 | • Single service organization (Direction de la biovigilance et de la biologie médicale (DBBM)) that provides oversight and resource planning across integrated testing environment. <br>• Single point of entry and somewhat transparent evaluation process for new tests through the DBBM and l'institut national d'excellence en santé et en services sociaux (health technology assessment program). <br>• Nimble financing approach with funding available for test development. | • Navigation, education, and coordination for care providers and patients are limited but in development. <br>• Limited integration of innovation and healthcare delivery. <br>• Limited engagement and involvement of broader stakeholder community, particularly commercial innovators. | 2 |

**Table 2.** *Cont.*

| Province | Population (M)/Area (1000 km²) | Testing Centres * | Strengths | Weaknesses | Rank ** |
|---|---|---|---|---|---|
| British Columbia | 5.0/944.7 | 4 | • Single service organization (Provincial Laboratory Medicine Services) that establishes a community of practice and supports resource planning.<br>• Single point of entry with explicit timelines for evaluation and coordination across service providers.<br>• Some integration of innovative testing. | • Lack of integration of laboratory information across centres.<br>• Limited engagement and involvement beyond care providers.<br>• Substantial opportunities to improve care navigation. | 3 |
| Nova Scotia | 0.97/55 | 2 | • Dedicated program (Pathology and Laboratory Medicine Program) that provides oversight and resource planning through key teaching hospitals.<br>• High level of service coordination.<br>• Integration of innovative testing. | • No single point of entry, explicit review process, timelines, or criteria used to consider new tests.<br>• Lack of integration of laboratory information across centres.<br>• Limited engagement and involvement beyond healthcare community. | |
| Ontario | 14.2/1076.4 | 19 | • Recently created single service organization (Provincial Genetics Program) intended to coordinate services and provide oversight and resource planning.<br>• Clear standards for accreditation and proficiency. | • Funding not timely or transparent; no funding for test development or human resources.<br>• No province-wide integration of laboratory information.<br>• Multiple evaluative frameworks, some not fit for purpose.<br>• Limited engagement and involvement beyond patients and care providers. | 4 |

* Each province additionally commissions out-of-province or out-of-country testing services; ** based on number of conditions established.

In Nova Scotia, a higher level of coordination and planning is achieved due to lower levels of service demand and the ability of the government to work directly with the individual teaching hospitals that provide province-wide testing. However, many of the operational and evaluative processes are informal, and not public facing. The opposite is true in Ontario, which is challenged with much higher service volumes relative to other provinces, a complex web of formal evaluative processes, and, until recently, a highly

decentralized health system. Ontario established a program dedicated to genetic testing in 2021, much later than Alberta and Quebec.

A further breakdown of the number of established conditions is shown in Figure 1. Sections 3.1.1–3.1.5 provide more detail, and a complete assessment of each province can be found in a larger report in Supplementary File S1.

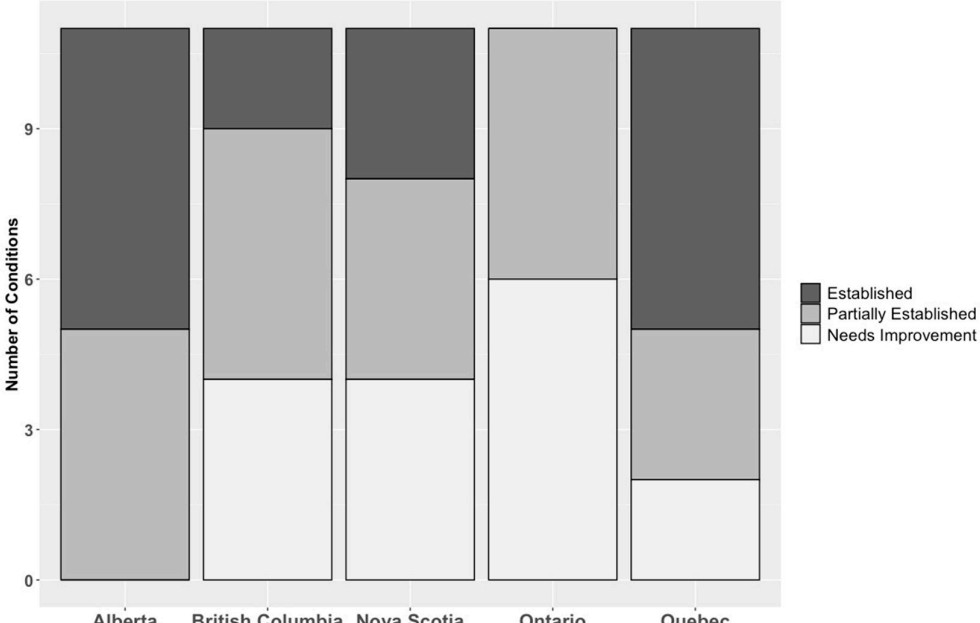

**Figure 1.** Number of necessary conditions for health system readiness for genome-based medicine established by province. Lightest shade indicates the condition does not exist or needs significant improvement ('needs improvement'). Intermediate shade indicates some aspects of the condition were established ('partially established'), while the darkest colour indicates most or all aspects of the necessary condition were established ('established').

### 3.1.1. Alberta

Alberta is Canada's fourth largest province by size and by population (approx. 4.2 million residents). The responsibility for testing is provided by a single organization, Alberta Precision Laboratories (APL), which is a wholly owned subsidiary of Alberta's single health authority, Alberta Health Services (AHS) [10]. Highly specialized genetic/genomic testing is delivered as a provincial program within APL, directly reporting to executive medical and operational leadership. Although the testing is operationalized at various locations, the test menu and coordination across sites of care are at a provincial level. Testing is also referred by the APL to out-of-province providers for rarer conditions.

*Infrastructure*

The APL provides an oversight function for the implementation of new tests on behalf of the AHS through established networks across care providers. Resource planning is conducted/coordinated by the APL. Alberta has worked toward a substantial integration of its electronic medical record systems across the province, creating a single integrated laboratory information system for testing.

*Operations*

The APL hosts a single-entry point for new testing. An intake form can be filled out by anyone (physicians, patients, innovators, or the public) [11]. The review process results in recommendations and advice being provided to the AHS regarding funding. The test review process (including a rapid Health Technology Prioritization Framework), timelines, and criteria used are under development, but not yet publicly available. The review process may also look at the decommissioning of tests. The APL works with the AHS to provide provincial coordination (e.g., for referral and sampling) for testing. A test

directory including navigational and supportive information is publicly available, although it does not provide information on all available testing (e.g., rare, genetic testing).

*Environment*

Alberta hosts a unique translational research program called the Health Innovation Platform Partnerships, which is aimed at small- to medium-sized enterprises. This has, in turn, led to the development of the Alberta Diagnostic Ecosystem Platform for Translation (ADEPT), hosted at the University of Alberta [12], to allow innovators access to clinical samples and related data to test, validate, and scale their technologies. Alberta also funds both well-established and investigational genetic tests for some conditions. The funds can be released by the AHS/APL for tests once an adoption decision is made, although tests with a large budgetary impact will require further consideration by the AHS. The funding formula for tests is not publicly available but is made after examining the costs associated with testing. The analytic standards are developed by discipline councils that work with individual hospitals and laboratories who are given the responsibility for testing through the APL. Alberta uses its own province-wide accreditation and proficiency standards based on the ISO 15189 standards.

Alberta has many of the necessary conditions and is currently leading Canada in its state of readiness for genomic medicine. Alberta requires a more transparent approach to create opportunities for innovation.

- **Established conditions**—Alberta has established conditions related to resource planning, informatics, a single-entry point for innovation, a coordinated service model, the integration of innovation and healthcare delivery, and regulation.
- **Partially established conditions**—Alberta still requires broader engagement with stakeholders beyond care providers and patients; a more transparent evaluation function; improved tools for care navigation; a more comprehensive and transparent finance approach that considers the associated costs of development, capital infrastructure, and human resources; and province-wide standards for education and training.
- **Conditions requiring significant improvement**—none.

### 3.1.2. British Columbia (BC)

Canada's second largest province by size and third largest province by population (approx. 5 million residents) has leveraged its single health authority dedicated to highly specialized services (the Provincial Health Services Authority (PHSA)) to coordinate the delivery of genetic testing. Highly specialized testing is delegated to larger teaching hospitals (Vancouver General Hospital; St. Paul's Hospital; Royal Columbian Hospital; BC Children's Hospital) depending on the type of test or therapeutic program. Testing is also referred to out-of-province providers for rarer conditions. The BC Provincial Laboratory Medicine Services (PLMS; formerly the BC Agency for Pathology and Laboratory Medicine) is the provincial program under the PHSA that is responsible for the administration and provision of insured laboratory benefits to British Columbians.

*Infrastructure*

The PHSA/PLMS provides an oversight function for the implementation of new tests on behalf of the BC Ministry of Health. The PLMS has established networks on its own and through the PHSA that are needed to understand the testing priorities and logistics of implementation. Resource planning is conducted/coordinated by the PLMS along with the PHSA. A unique challenge in BC is that individual regions and hospitals have separate governance structures and do not have integrated laboratory information systems.

*Operations*

A single-entry point for new testing is provided through the PLMS test review process. The process is open only to employees and/or authorized agents of a laboratory provider, a health authority, and the BC's Agency for Pathology and Laboratory Medicine or Ministry of Health and can have support from a co-applying physician. The review process results in recommendations and advice being provided to the Ministry of Health regarding funding. While the criteria for the test review process are made publicly available, the review process

and rationale for the test recommendations are not. The service coordination for testing is provided by the PLMS, although the regional coordination (e.g., for referral and sampling) is carried out by individual health authorities. There are currently no published test lists or protocols for care providers, and the navigation for the access to testing is limited to private communication with specialist providers/centres, and the use of healthcare navigators for some care programs.

*Environment*

BC has a strong translational research environment with the BC Michael Smith Genome Science Centre (GSC), which receives funding for sequencers and acts as a research arm of the PHSA through its accredited laboratory. BC funds both well-established and investigational genetic tests for some conditions. The funds can be released for tests once an adoption decision is made; for companion diagnostics in cancer, the funding is provided through the cancer drug budget. The funding formula for tests is based on traditional community-based testing and requires revision. Additional funding for more complex testing can be released at the Ministry's discretion. There are no province-wide standards for education and training related to testing. Accreditation and proficiency are governed through a province-wide accreditation standard similar to the College of American Pathologists (the Diagnostic Accreditation Program (DAP), ISO 15189). The analytic standards are developed by individual hospitals and laboratories who are given responsibility for testing through the PLMS.

British Columbia is taking the necessary steps to advance its system readiness for genomic medicine. Some challenges relate to its decentralized health system structure (informatics, navigation, province-wide standards), while others may be more easily remedied.

- **Established conditions**—British Columbia has established conditions related to resource planning and regulation.
- **Partially established conditions**—British Columbia still requires broader engagement with stakeholders beyond the clinical community, an open application process for innovation proposals, a more coordinated service model, further integration of innovation and healthcare delivery, and a more comprehensive finance approach.
- **Conditions requiring significant improvement**—British Columbia has yet to establish linked laboratory information systems, a transparent evaluation function, improved tools for care navigation, and province-wide standards for education and training.

### 3.1.3. Ontario

Ontario is the largest of Canada's 13 provinces and territories by population (approx. 14.8 million residents, with the vast majority of the province's inhabitants located in its southernmost regions), and the third-largest province by size. Its capacity for genetic testing largely resides in its hospitals, with testing for hereditary diseases largely occurring in Ontario's two children's hospitals. Some testing is commissioned to out-of-province providers as well. Somatic testing is conducted across 11 centres of varying sizes.

*Infrastructure*

The networks of genetic testing providers were originally established through the Regional Genetics Program, the Ontario Molecular Pathology Research Network, tumour site groups, and other clinical programs, such as the Pathology and Laboratory Medicine Program (PLMP). Overarching coordination has now become the responsibility of the newly (2021) established Ontario Health Provincial Genetics Program (PGP), which may utilize these networks or create new ones. Resource planning for genetics is still conducted on a hospital or regional level through regional bodies of Ontario Health as Ontario moves toward a more centralized model of care delivery. Working groups were established in 2017 to examine the health human resources required for clinical genetic services, and this, along with data and digital systems, is now a focus of the newly established PGP.

*Operations*

There is no single-entry point for new testing. The current Ontario Health (Quality, OHQ) process allows manufacturers of commercial innovation and translational researchers to apply for assessment; however, the priorities for what ultimately gets assessed is assigned to the OHQ [13]. Beyond this, proposals for new testing rely on practicing clinicians and other internal stakeholders (such as pharmacy services). The tests may be evaluated through several routes including the Program in Evidence-Based Care [14]/tumor site groups, the Ontario (Quality)/Ontario Genetics Advisory Committee, the Ontario Public Drug Programs/the Ontario Steering Committee for Cancer Drugs (OSCCD), and Newborn Screening Ontario/the Newborn Screening Ontario Advisory Council (NSO-AC) [15]; each uses different evaluative frameworks. Coordination across laboratories is achieved through the PGP for hereditary and somatic testing. There is a defined test list for both hereditary and somatic testing; navigation is largely provided by genetic testing centres, specialty clinics, or patient organizations, and the PGP is planning to consolidate these navigational resources for patients and providers.

*Environment*

In May 2021, Ontario committed to implementing comprehensive cancer testing including genetic panels that include both established and investigational tests. There are also implementation pilot projects such as the Genome-wide Sequencing Ontario (GSO), which is aimed at providing genome-wide sequencing to 2000 Ontario families [16]. Ontario still largely relies on annual budget cycles and Ministry decisions to release funding for new tests. Accreditation and proficiency are based on the ISO 15189 medical laboratories standard.

Ontario has only recently taken the necessary steps to improve its state of readiness for genomic medicine. However, it currently lacks many of the necessary conditions to be prepared.

- **Established conditions**—Ontario has not yet fully established any of the necessary conditions required for health system readiness.
- **Partially established conditions**—Ontario still requires broader engagement with stakeholders beyond care providers and patients, ongoing resource planning, a more coordinated service model, better integration of innovation with mainstream healthcare delivery, and province-wide analytic standards (regulation).
- **Conditions requiring improvement**—Ontario has yet to establish linked laboratory information systems; create a single-entry point for innovation; a single, fit-for-purpose evaluation function; tools for care navigation; province-wide standards for education and training; and a more comprehensive finance approach. Some of these are currently being planned for development.

### 3.1.4. Nova Scotia

While Nova Scotia has a population of less than 1 million (less than 3% of Canada's population), it is the most populous province in Canada's Atlantic region. Testing occurs within two major hospitals (Queen Elizabeth II Health Sciences Centre and IWK Health Centre) that deliver specialized care programs. Nova Scotia also uses out-of-province providers. Oversight for these programs is provided by the Nova Scotia Health Authority (NSHA) through its Pathology and Laboratory Medicine Program (PLMP).

*Infrastructure*

The small population size of Nova Scotia has likely negated the need for large-scale healthcare networks. Communities of practice are established within specialized care programs as well as the Pathology and Laboratory Medicine Program. The oversight function for the implementation of new tests is the responsibility of the individual hospitals, who are, in turn, given a budget to deliver specialized programs of care. Analytic standards, as well as resource planning, are developed by these individual hospitals. A single integrated laboratory information system for testing does not exist.

*Operations*

There is no single-entry point for new testing. The test review is conducted through a provincial advisory committee, but the process, timelines, and criteria used are not publicly available. A Laboratory Test Catalogue including navigational and supportive information is publicly available.

*Environment*

Nova Scotia provides access to both well-established and investigational genetic tests for some conditions. Funding mechanisms are hindered by the reliance on hospital budgets and annual budget cycles. The funding formula for tests is not clear. There are no province-wide standards for education and training related to testing. Nova Scotia uses the Canadian Counsel on Health Service Accreditation (CCHSA) province-wide accreditation standards based on the ISO 15189 standards. Proficiency testing is voluntary.

Nova Scotia's state of readiness for genomic medicine is aided by its size and established teaching hospitals. However, many of its processes are unclear. Nova Scotia would benefit from further integration and engagement with the broader innovation community.

- **Established conditions**—Nova Scotia has established conditions related to resource planning, a coordinated service model, and the integration of innovation with mainstream healthcare delivery.
- **Partially established conditions**—Nova Scotia still requires broader engagement with innovation stakeholders, an improved financing approach, and province-wide analytic standards (regulation). Nova Scotia has taken some steps to aid care navigation (i.e., test directory and ongoing communication to providers), but not all tests (e.g., oncology) are listed.
- **Conditions requiring improvement**—Nova Scotia has yet to establish linked laboratory information systems; create a single-entry point for innovation; a single, fit-for-purpose evaluation function; and standards for education and training.

3.1.5. Quebec

Canada's largest province by size and second largest by population (approx. 8.5 million residents) began the reform of its system of laboratory governance in 2011. Molecular diagnostics including low- to medium-throughput sequencing is delivered across five 'clusters' operating seven supra-regional laboratories (Capitale-Nationale (CHU de Québec—Université Laval); Estrie (CHUS—Hôpital Fleurimont); Montréal—CHUM (CHUM and Hôpital Maisonneuve-Rosemont); Montréal—CUSM (CUSM and Hôpital général Juif); Montréal—CHU Sainte-Justine (CHU Sainte-Justine)) as well as the Montreal Heart Institute (MHI). The Centre québécois de génomique clinique (CQGC) in 2018, physically situated at the Centre hospitalier universitaire Sainte-Justine (CHU Sainte-Justine), was established to conduct high-throughput (exome, transcriptome, or whole-genome) sequencing. Testing is also referred to out-of-province providers for rarer conditions. The Direction de la biovigilance et de la biologie médicale (DBBM) is the ministry program that has been tasked with coordinating the implementation of molecular diagnostic testing across all of these centres/clusters [17].

*Infrastructure*

Under the DBBM, the Réseau québécois de diagnostic moléculaire (RQDM) acts as the supra-regional network and advisory function for the DBBM on behalf of the Quebec Ministry of Health. Resource planning for genetics is conducted by the seven centres/clusters (five regions, MHI, and CQGC) and is coordinated by the DBBM. Further coordination across oncology centres (through the Quebec cancer program (PQC)) is now underway, given a recognized need for further coordination in cancer program delivery. A laboratory information system is being established across this network.

*Operations*

The DBBM acts as an entry point for new testing. The DBBM works with L'institut national d'excellence en santé et en services sociaux (INESSS) [18] to provide advice to the Ministry of Health regarding funding. Only public laboratories can submit requests

to the DBBM for evaluation by INESSS. In the case of companion diagnostic tests, drug manufacturers must submit the diagnostic test evaluation request with the drug evaluation request. Tests are evaluated through a single evaluative framework, and recommendations to the Ministry are made public, although there is a limited engagement with the stakeholders in this evaluation process. While there is a test formulary (the Répertoire québécois et système de mesure des procédures de biologie médicale), it may not always be clear how and where a test can be made available to patients. The RQDM is currently working on providing additional navigational support.

*Environment*

Quebec has continued with a policy of only funding medically necessary tests and will not pursue the reporting of investigational tests that are associated with unfunded targeted therapies. Translational research projects are conducted through the CQGC and Genome Quebec is asked to participate in the development and validation of standard operating procedures for high-throughput testing. The funds can be released by the Ministry once an adoption decision is made, and funding for test development and additional human resource costs has more recently (2021) been provided. It is unclear whether the funding formula is value-based or amenable to reassessment. The DBBM/RQDM has committed to developing province-wide standards for education and training. Accreditation and proficiency are regulated through a province-wide accreditation standard (ISO 15189). There are no province-wide analytic standards.

Quebec began taking the necessary steps to reform its approach to genome-based testing over a decade ago. There are still opportunities to improve the optimal use of testing in Quebec.

- **Established conditions**—Through the DBBM, Quebec has established conditions related to resource planning, a more robust finance approach, and standards for analysis, accreditation, and proficiency (regulation). It also has an established evaluation function (with INESSS) and linked information systems.
- **Partially established conditions**—Quebec still requires broader engagement with stakeholders beyond care providers. While it has a single-entry point for innovation, it is closed to commercial innovators. It similarly lacks standards for education and training, but these are being developed. While it has created a coordinated service model, there are still further opportunities to improve the coordination and timing of testing.
- **Conditions requiring improvement**—Quebec still does not fully integrate investigational testing into mainstream care. Awareness and navigational tools for care providers and patients is lacking; not all available tests are published on its test list ('repertoire').

## 4. Discussion

Overall, these findings suggest that Canada has created some of the established conditions, but more action needs to be taken to improve the state of readiness for genome-based medicine. The uneven development across provinces further highlights the challenges of federated (pluralistic) health systems. Canada's provinces and territories are all individually responsible for healthcare delivery and harbor different governance and information structures. While some Canadian provinces are currently taking steps to improve the delivery of genetic services, there may be considerable hurdles for them to overcome. Creating conditions related to linked information systems and coordination of care may be more difficult in larger provinces and in those with more decentralized systems of care. For some provinces, this means that considerable time and investments are needed to further develop the necessary conditions for readiness.

These findings also highlight the roles of organizational context, social influence, and other factors that are known to affect the diffusion of innovation within health systems [19,20]. Genetic testing is a heavily context-dependent (i.e., complex) intervention that is reliant on care pathways, referral patterns, organizational levels, education and training, and other

interacting components [4]. The health system change toward genomic medicine may be motivated by a number of different factors including an increasing number of specialists reliant on testing, laboratory leaders, and innovation programs. One interviewee speculated that the lack of motivation for some healthcare administrators might best be explained by relatively small expenditures compared to the overall spend. In the province of Ontario, for example, laboratory services represented 4% of the total healthcare expenditure in 2016, and genetic testing only represents 5% of these costs (i.e., 0.2% of all healthcare costs) [21].

While this may be true, our findings suggest that the provinces at a higher state of readiness were the ones who made it a political priority to consolidate laboratory services into single service organizations at an earlier stage. These single service organizations can, in turn, foster many of the conditions required for readiness including the following: (1) creating communities of practice and regional networks; (2) systematic oversight and resource planning; (3) single points of entry for new testing proposals; (4) an evaluation function; (5) coordination of service; (6) tools for awareness and care navigation; (7) standards for education and training; and (8) regulatory standards related to analysis. Some remaining conditions may be beyond the scope of a laboratory service (or any single health service) organization including informatics, how innovation is integrated into healthcare delivery, and approaches to financing.

The consolidation of laboratory services into a single service organization may be greatly facilitated by already-centralized healthcare environments such as those in Alberta and Quebec. International examples can also be seen and include both the UK and US. The NHS England was able to reorganize its existing capacity in 2018, creating a Genomic Medicine Service through its Genomic Laboratory Hubs, each hosted by an acute NHS trust and designated a geographic region for coverage [22]. Similarly, the US Department of Veteran's Affairs has leveraged its existing capacity to deliver genetic testing through its oncology program and dedicated service centres across the US toward non-oncologic indications for testing [23]. The latter model (service coordination through an oncology program) is now being developed in Ontario.

Our findings also shed a light on the emergence of health technology assessment (HTA) as a policy tool to address test decisions. While its use in each province is promising, the HTA processes for testing in Canada are still generally lacking in key HTA principles [24,25] and good HTA practices [26]. This includes processes that are consistent, timely, transparent, and responsive to and engaged with stakeholders [25,27]. A more open and engaged approach to technology management may be foreign to laboratory leaders and Canadian health administrators who have, until now, managed laboratory technology using smaller internal processes. However, the interplay of societal preferences reflected in genome-based testing, including the need for equitable healthcare and the potential for a large number of technology proposals and unmanageable expenditure growth, necessitates a 21st century approach to HTA.

Our findings also shed some light on the challenges with financing genetic testing services. The responsibility for genetic testing in Canada has largely fallen to hospital laboratories that are, in turn, funded through provincial block funding arrangements and annual budget cycles, along with public and private research grants and private fundraising. These annual funding envelopes give hospitals the ability to quickly adopt and deliver new technology when they are seen as medically necessary and affordable. However, when the human resource, capital, and operational costs of genetic testing are seen as too high, laboratory leaders must rely on additional provincial funding, which can be slower than the speed of innovation. To date, many provinces have used research funding, often from drug companies with targeted therapies requiring testing, to make up for a shortfall in funding for the development and delivery of tests. This can lead to substantive problems with governance and fiscal management where public sector actors are highly dependent on the private sector for the delivery of public services, and yet, public actors remain accountable to the public at large.

Adding to this challenge is a need to change the funding formula for genetic testing. The charges for community-based testing were traditionally operationalized through test schedules and based on historical costs of labour consumables and caps to limit excessive expenditure. The current per-test price that is constrained to test type, approach, and patient type also does not consider the efficiencies that could be realized with changes in the testing type (e.g., multigene assay versus single gene approaches), approach (e.g., reflex testing or upfront testing versus ordered testing or sequential testing), or patient type (first line versus second line). Genetic tests are much more costly in terms of consumables and labour and require a considerable upfront investment to implement, challenging these traditional per-test costing assumptions. While the funding formula for new tests was not publicized in most provinces, it appears that among this survey of provinces, only Quebec has made changes to its financing approach to account for some of this shortfall.

While these findings are accurate as of this publication, they should be considered a snapshot in time. Further to this, some healthcare system planning is not publicly disclosed. While health system leaders were approached and welcome to comment on these findings, there may have been some aspects of care under development that could not be discussed or shared. Another limitation is that this report does not represent a survey of Canada as a whole. An informal discussion with administrators in smaller provinces (e.g., Newfoundland, Manitoba, Saskatchewan) suggested that there is a much greater reliance on out-of-province sourcing for testing in these areas, including referrals to other provinces or with commercial vendors in Canada, the US, or Europe.

Despite this, we believe the conditions previously developed are universal and could be applied to other healthcare systems to gauge system readiness. They may also be applicable to other advanced forms of testing that continue to emerge, and may be used in parallel with genomic information (e.g., metabolomics, proteomics). Testing, while already commonplace in cancer diagnosis, prognosis, and treatment, will become increasingly prevalent in a number of disease areas including the following: the diagnosis and treatment of rare diseases, autoimmune diseases, cardiology, psychiatry, ophthalmologic conditions, lung diseases, infectious diseases, neurology, and in the use of cell and gene therapies. Many of these represent considerable healthcare expenditure today. We found that some provinces in Canada have more developed genetic services focused on pediatric diagnosis and oncology but may not be prepared for these emerging disease areas. In British Columbia, for example, funds can be released quickly for targeted treatments in cancer, but this same funding arrangement does not exist for non-cancer medicines.

## 5. Conclusions

These findings suggest that Canada's major healthcare regions are moving toward a state of readiness for genomic medicine, although they are using different approaches and at different rates. They highlight the many challenges that health systems face when they are required to quickly respond to a disruptive technology. Even more so, these findings highlight the differences in the access to care that Canadians may face when they are served by individual health regions with different priorities and healthcare structures. Even if change is desirable, health systems need to be designed to be responsive and resilient to be able to quickly shift priorities and recognize and enable valuable innovation [28].

**Supplementary Materials:** The following supporting information can be downloaded at https://www.mdpi.com/article/10.3390/curroncol30060408/s1, File S1: Full Report.

**Author Contributions:** All authors (D.H., E.V., V.M., M.M., C.I., L.S., D.S.S., B.S., S.Y., P.J., T.S. and L.A.) contributed to the conceptualization, methodology, and writing—review and editing of this paper and informed the list of conditions. D.H. was responsible for the funding acquisition and led the writing—original draft. All authors approved the final version of the article. D.H. is the guarantor of this work. The corresponding author attests that all listed authors meet the authorship criteria and that no others meeting the criteria have been omitted. The conclusions of the authors were not contingent on the sponsor's approval or censorship of the manuscript. The conclusions are drawn by the authors and do not reflect the views of their affiliated organizations. All authors have read and agreed to the published version of the manuscript.

**Funding:** Funding for this study was provided by the following: Amgen Canada, Inc.; AstraZeneca Canada Inc.; Eli Lilly Canada, Inc.; GlaxoSmithKline Inc. (GSK Canada); Hoffmann-La Roche Canada, Inc. (Diagnostics Division); Janssen/J&J Canada Inc.; Pfizer Canada ULC; Thermo Fisher Scientific Inc. All sponsors contributed equally. None of the sponsors played a role in drafting, revising, or approving the content of this research.

**Conflicts of Interest:** The authors declare the following conflicts of interest. D.H. and L.S. report funding, outside the submitted work, from various pharmaceutical companies with an interest in genome-based testing and precision medicine. D.S.S. is a salaried employee of Menarini Silicon Biosystems, Inc. (Huntingdon Valley, PA, USA) which is a commercial diagnostic manufacturer/clinical laboratory. V.M. advises a range of companies, including those with products and services related to the subject of this paper. V.M. holds, and has held shares in health data, health services, healthcare consultancy, and biotech companies, and serves or has served in paid non-executive director roles and paid advisory roles in such companies. The funders had no role in the design of the study; in the collection, analyses, or interpretation of data; in the writing of the manuscript; or in the decision to publish the results.

## Appendix A. Interview Guide

*Background*

I have been asked by a consortium of companies (Amgen Canada Inc., AstraZeneca Canada, Eli Lilly Inc., GlaxoSmithKline Inc. (GSK Canada), Janssen Inc./J&J, Pfizer Canada ULC., Thermo Fisher Scientific Inc., and Roche Canada) to investigate what the current and future state of readiness for advanced (genome-based) diagnostic testing in Canada is and might become.

By advanced diagnostic testing we mean molecular testing (DNA testing such as sequencing, PCR and DNA microarray), cytogenetics (chromosome testing such as karyotyping and FISH), and testing for metabolic products (protein testing through immunoassay or immunohistochemistry).

This work is to help all involved in advanced diagnostic testing in Canada to identify what can be done to make sure that Canadian health systems are prepared for the future of testing.

You have been identified as someone with expert knowledge in the area who could provide significant value to understanding the present and future of advanced diagnostic systems either nationally and internationally. As such, we would like to discuss the subject with you by phone for 45 to 60 min in a semi-structured interview.

This interview would cover your specific areas of expertise and the content developed through this interview would help inform the creation of a report that will be made available publicly.

Your contribution to this report will be acknowledged as a key informant, but there will be no comments specifically attributed to you. Notes from the interview will be shared with you after the call to ensure accuracy and to identify any areas of clarification required

*Semi-structured interview guide*

The interview begins with the interviewer stating the purpose of the interview, the topics that he wants to explore and the depth of response expected [29].

*Purpose:*

Interviewer: The purpose of today's interview is two-fold:

1. It will help identify current challenges with the uptake and routine delivery of advanced diagnostic testing
2. To explore what conditions are necessary and desirable for creating robust systems of advanced diagnostic testing (either in your region or generally)

Interviewer: I would like to cover a few topics today that will help answer the question concerning how advanced testing is being conducted today and what changes may be necessary to ensure its continued and effective delivery.

In particular I would like to explore your views on what approaches to the introduction and evaluation of tests, their development and validation, and financing of tests are needed as well as human resource and infrastructure required.

In each case, I will try to describe how much feedback is needed. However, I want to encourage you to speak freely in response to each question, even if you feel it doesn't directly address the question. We will have [time] for discussion.

*Questions*

1. Do you feel the current testing services offered are sufficient to keep up with the current and future demand for advanced testing?
2. What are the current challenges with the uptake and routine delivery of advanced diagnostic testing?
   - Do you have any cases that exemplify these challenges?
   - Do these challenges differ depending on whether testing is intended for diagnosis, therapeutic decisions, or hereditary testing?
3. What do you feel needs to change in order to keep up with current/future demand and address these challenges?
   - Who are the key decision makers, organizers and administrators of advanced testing that are currently involved?
   - Who else needs to be involved?
   - Are there any proposed changes currently?
4. Do you have any further thoughts on what needs to change to support a more nimble approach to the awareness, acceptance, and adoption of advanced testing?
5. Permission to Use Name, Interviewee demographics.

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
