# Peer review of "Progress toward Health System Readiness for Genome-Based Testing in Canada"

_curroncol, doi:10.3390/curroncol30060408_

Round 1

Reviewer 1 Report

The manuscript focuses on the identification of logistic strategy adopted in Canadian regions to performic genomic tests for predictive molecular analysis. In my opinion, the manuscript is  technically correct but requires some moderate implementations to accept this paper on this journal

- In the methodological section, please, could the authors consider other data that may play a pivotal role in the strategic approach adopted to perform molecular tests? Accordingly, I would reccomend to consider other parameters including number of tested biomarkers, turaround time (TAT) and cost/sample. As showed by previous published literature data, (PMID: 34813925) these data may play a curcial role to carry out molecular analysis.

- In addition, I owuld suggest to describe the biological source adopted by each area for molecular tests. Overall, I would reccomend to implment other technical considerations about testing startegy.

- Please, could the authors improve the quality of figure? Moreover, could the authros add a table sumamrizing these findings?

Fine minor english modifications should be approached

Author Response

The manuscript focuses on the identification of logistic strategy adopted in Canadian regions to performic genomic tests for predictive molecular analysis. In my opinion, the manuscript is  technically correct but requires some moderate implementations to accept this paper on this journal

- In the methodological section, please, could the authors consider other data that may play a pivotal role in the strategic approach adopted to perform molecular tests? Accordingly, I would reccomend to consider other parameters including number of tested biomarkers, turaround time (TAT) and cost/sample. As showed by previous published literature data, (PMID: 34813925) these data may play a curcial role to carry out molecular analysis.

- In addition, I owuld suggest to describe the biological source adopted by each area for molecular tests. Overall, I would reccomend to implment other technical considerations about testing startegy.

Response: We thank the reviewer for the comment but are not sure how these parameters should be best considered. Volumes of tests, turnaround time and cost per sample are certainly important metrics to consider when adopting tests but our current work does not focus on what factors should be considered when adopting tests. Instead, it considers factors that are required to be prepared to adopt tests. To allay any confusion, we have called attention to this in the introduction section (and describing these as conditions for preparedness rather than conditions for delivering effective/efficient care)

- Please, could the authors improve the quality of figure? Moreover, could the authros add a table sumamrizing these findings?

Response: We’ve replaced the figure with a table

Reviewer 2 Report

In this manuscript, the authors collect and summarize expert opinions about the status of genetic testing in Canada by assessing a previously published list of healthcare readiness conditions. This was well written and an interesting read, and an important summary of current challenges facing Canadian healthcare systems.

I was especially interested in the comparison to the UK’s health care system, and would also like to read some discussion/comparison to the US health care system, which is also heavily involved in genetic testing. Some comparisons to other countries’ systems and/or challenges would be of much interest to the international scientific community.

The manuscript mentioned Figure S1, Table S1, Video S1, and Appendix B, but these files are not available for review. Please provide. Other than this, some minor comments and questions below:

line 127: "The Canadian provinces in a better state of readiness for genomic medicine are Alberta,  Quebec and British Columbia." - However, based on Fig 1, Ontario and Nova Scotia appear equally ready. Please edit the text or add in one more shade of orange to distinguish.

Fig1 figure resolution is very low and the text is barely readable.

While the "conditions" are mentioned multiple times, and the authors' previous paper is cited, they are not actually introduced in any way. It would be an easier read if they were briefly introduced/listed in the introduction section.

Fig 2- “No fill indicates the condition does not exist or needs significant improvement.” This is confusing because I don’t see any that are “no fill”

The term ‘funding formula’ is used throughout. Please define or be more specific in describing this term.

How is the CCMG involved in standardized education and accreditation across the provinces in Canada?

Typos:

-line 29 "methodsapproach", line 86 "health Ministry"

-some reference numbers are provided before periods and some after.

- line 171, space before comma

Some of the conditions aren’t very specific as to what the authors mean. For example all four provinces discussed “require[s] broader engagement 198 with innovation stakeholders”

Line 135 mentions Ontario has “much higher levels of demand for service” but is referred in lines 448-452 as “small numbers” and only “0.2% of healthcare costs.” If so, other provinces may have even smaller numbers?

Lines 517-518: “We found that some provinces in Canada have more developed genetic services focused on pediatric diagnosis and oncology but may not be prepared for these disease areas.” Please clarify what ‘these’ refers to -> I assume the authors are referring to the rare disease listed in lines 515-516, but it is not clear in this sentence.

Author Response

In this manuscript, the authors collect and summarize expert opinions about the status of genetic testing in Canada by assessing a previously published list of healthcare readiness conditions. This was well written and an interesting read, and an important summary of current challenges facing Canadian healthcare systems.

I was especially interested in the comparison to the UK’s health care system, and would also like to read some discussion/comparison to the US health care system, which is also heavily involved in genetic testing. Some comparisons to other countries’ systems and/or challenges would be of much interest to the international scientific community.

Response: We thank the reviewer for the comment and agree that readers might be interested in better contextualizing results. We have added a paragraph in the discussion section accordingly, that attempts to discuss provide more contrasts between the UK and US and highlights the strength of using a single, testing-service organization.

The manuscript mentioned Figure S1, Table S1, Video S1, and Appendix B, but these files are not available for review. Please provide. Other than this, some minor comments and questions below:

Response: This was a cut and paste error. Only two supplementary pieces of material have been referred to and these have been attached. (Appendix 1 – Semi-structured interview guide; Appendix B- larger report)

line 127: "The Canadian provinces in a better state of readiness for genomic medicine are Alberta,  Quebec and British Columbia." - However, based on Fig 1, Ontario and Nova Scotia appear equally ready. Please edit the text or add in one more shade of orange to distinguish.

Fig1 figure resolution is very low and the text is barely readable.

Response: To deal with these concerns , the figure has been replaced with a table.

While the "conditions" are mentioned multiple times, and the authors' previous paper is cited, they are not actually introduced in any way. It would be an easier read if they were briefly introduced/listed in the introduction section.

Response: We agree and have introduced these in the introduction with some description of what they mean and why they are needed.

Fig 2- “No fill indicates the condition does not exist or needs significant improvement.” This is confusing because I don’t see any that are “no fill”

Response: Have changed this to “the lightest shade” rather than “no fill” to avoid confusion as well as added some further descriptors.

The term ‘funding formula’ is used throughout. Please define or be more specific in describing this term.

Response: This term has been defined in the introductory table “(i.e., payment that accounts for development and human resource costs and benefits)” of conditions and again in the text at the first instance of its use “(i.e., payment model for the reimbursement of test services)”.

How is the CCMG involved in standardized education and accreditation across the provinces in Canada?

Response: They certainly offer training and accreditation, but our focus was on province-wide mandates . There are other examples of accreditation programs offered but considered voluntary or not mandated within provinces.

Typos:

-line 29 "methodsapproach", line 86 "health Ministry"

-some reference numbers are provided before periods and some after.

- line 171, space before comma

Response: Corrected.

Some of the conditions aren’t very specific as to what the authors mean. For example all four provinces discussed “require[s] broader engagement 198 with innovation stakeholders”

Response: We have provided more nuanced and tailored messages per province; some provinces (such as Alberta) do engage with care providers and patients; others engage only with care providers. The table further provides a good practice description and what is meant by broad engagement “Engagement with all stakeholders, including administrators, information and communications technology professionals, implementation and genome scientists, and public and private sector innovators.”. This was already mentioned in the first paragraph of results.

Line 135 mentions Ontario has “much higher levels of demand for service” but is referred in lines 448-452 as “small numbers” and only “0.2% of healthcare costs.” If so, other provinces may have even smaller numbers?

Response: We don’t think thesefacts  are contradictory. Ontario has much higher levels of service (have changed to “service volumes relative to other provinces”) even though expenses are a small proportion of its budget. Other provinces would have similar proportional expenditures but lower volumes.

Lines 517-518: “We found that some provinces in Canada have more developed genetic services focused on pediatric diagnosis and oncology but may not be prepared for these disease areas.” Please clarify what ‘these’ refers to -> I assume the authors are referring to the rare disease listed in lines 515-516, but it is not clear in this sentence.

Response: We have corrected the paragraph to make it clearer to the reader.